# Health care professionals' views towards self-management and self-management education for people with type 2 diabetes

Jamie Ross,[1] Fiona A Stevenson,[1] Charlotte Dack,[2] Kingshuk Pal,[1] Carl R May,[3] Susan Michie,[4] Lucy Yardley,[5] Elizabeth Murray[1]

¹Department of Primary Care & Population Health, University College London, London, UK
²Department of Psychology, University of Bath, Bath, UK
³London School of Hygiene and Tropical Medicine Faculty of Epidemiology and Population Health, London, London, UK
⁴Centre for Outcomes Research and Effectiveness, University College London, London, UK
⁵Nuffield Department of Primary Health Care Sciences, University of Oxford, Oxford, UK

**Correspondence to**
Professor Elizabeth Murray;
elizabeth.murray@ucl.ac.uk

## ABSTRACT

**Objectives** Significant problems with patients engaging with diabetes self-management education (DSME) exist. The role of healthcare professionals (HCPs) has been highlighted, with a lack of enthusiasm, inadequate information provision and poor promotion of available programmes all cited as affecting patients' decisions to attend. However, little is known about HCPs' views towards DSME. This study investigates the views of HCPs towards self-management generally and self-management in the context of DSME more specifically.

**Design** A qualitative study using semi-structured interviews to investigate HCPs views of type 2 diabetes self-management and DSME. Data were analysed thematically and emergent themes were mapped on to the constructs of Normalisation Process Theory (NPT).

**Setting** Two boroughs in London, UK.

**Participants** Sampling was purposive to recruit a diverse range of professional roles including GPs, practice nurses, diabetes specialist nurses, healthcare assistants (HCAs), receptionists and commissioners of care.

**Results** Interviews were conducted with 22 participants. The NPT analysis demonstrated that while a self-management approach to diabetes care was viewed by HCPs as necessary and, in principle, valuable, the reality is much more complex. HCPs expressed ambivalence about pushing certain patients into self-managing, preferring to retain responsibility. There was a lack of awareness among HCPs about the content of DSME and benefits to patients. Commitment to and engagement with DSME was tempered by concerns about suitability for some patients. There was little evidence of communication between providers of group-based DSME and HCPs or of HCPs engaging in work to follow-up non-attenders.

**Conclusions** HCPs have concerns about the appropriateness of DSME for all patients and discussed challenges to engaging with and performing the tasks required to embed the approach within practice. DSME, as a means of supporting self-management, was considered important in theory, but there was little evidence of HCPs seeing their role as extending beyond providing referrals.

## INTRODUCTION

Self-management has been characterised as a key feature of contemporary healthcare

### Strengths and limitations of this study

► One of the first studies to explore healthcare professionals' (HCPs') views towards type 2 diabetes self-management education (DSME) in routine practice.
► A wide range of HCPs were interviewed, including GPs, nurses, commissioners, healthcare assistants (HCAs) and administrative staff.
► Participants had varying degrees of prior knowledge and experience of DSME.
► A theoretical framework Normalisation Process Theory was used to analyse the findings.
► The sample was limited to HCPs within two London boroughs.

systems.[1] Supporting self-management by patients with chronic conditions is now an accepted and important part of reducing disease burden and health service use associated with chronic disease in many countries.[2][3] For diabetes, self-management education (DSME) offers strategies to offset the challenges that providers face in delivering chronic disease care, while also improving outcomes for patients.[4] Globally, however, there are serious problems with patients with diabetes participating in DSME. Research from the UK,[5–7] USA,[8] Mexico,[9] Germany,[10] France,[9] Italy,[9] India,[11] Russia,[9] Algeria,[9] Turkey,[9] China[9] and Canada[12] report low rates of patient attendance.[13] In the UK, DSME is recommended for people with type 2 diabetes (T2DM),[14] and primary care services (GP practices) are financially incentivised to refer patients to available programmes.[15] However, data from the UK's National Audit Office survey suggests that in 2015 only 8.2% of patients with T2DM attended DSME.[16] Poor attendance rates are a major concern given that high quality DSME can have positive effects on quality

of life and health outcomes[17–20] and that patients who do not attend any form of diabetes educational intervention are at a fourfold increased risk of developing complications.[21]

In the UK, problems previously existed with health professionals not referring patients to structured education. However referrals to DSME were made a Quality and Outcomes Framework (QOF) indicator, resulting in payment incentives to GP practices for referral in 2013/4.[22] This increased rates of referral from 15.9% in 2012/2013 to 75.8% in 2014/2015. However, the rate of patient uptake remained low, only increasing from 3.6% in 2012–13% to 5.3% in 2014–2015,[23] representing a problem with translating the referrals into attendance.

Research on reasons for non-attendance at DSME suggests there are factors which relate to patients being unable to attend (eg, because of accessibility issues, physical health problems and financial problems) and others that relate to patients choosing not to attend (eg, because of a lack of perceived benefit, knowledge or information or because of emotional and cultural factors).[6] A number of strategies are suggested to overcome barriers identified by patients, including the provision of more culturally specific education,[24] the use of alternative methods of delivery, such as online[17] and better promotion of education by health professionals.[6 24] The role of healthcare professionals (HCPs) as pivotal in patient decisions to attend DSME has been highlighted, with a lack of enthusiasm, inadequate information provision and poor promotion of available programmes by HCPs all cited as affecting patients' decisions to attend DSME.[6 7 11 12 25 26] Research has also found variation in terms of HCPs' level of knowledge about diabetes education[24] which is potentially important given the important role they play in encouraging patients to attend.

Despite HCPs potentially playing a key role in integrating DSME into routine care delivery[4] and promoting self-management,[27] there is little research into the views of HCPs towards DSME. One paper exploring HCPs' views towards group-based DSME focused largely on practice nurses who were knowledgeable about DSME having either been educators, or attended a taster session of group-based DSME.[24] These practice nurses viewed DSME favourably, particularly the group mode of delivery, reporting that it improved patient interactions saving HCPs' time and improved patient outcomes. However, they also reported that DSME was not accessible to those with literacy problems, older people and those who worked or had young children Other research suggests that HCPs may be ambivalent about the importance and benefits of self-management support programmes for chronic illnesses, and are concerned about sharing responsibility for disease management with other professional educators or even patients themselves.[27] It has also been suggested that if HCPs perceive these self-management programmes to be ineffective or inaccessible for their patients they may be less likely to employ these resources for their patients.[4] Furthermore there has been little research into HCPs' views on alternative forms of DSME such as online.[28]

The aim of the current study was to explore the views of HCPs towards a self-management approach to diabetes care for patients with T2DM within two socially and economically diverse settings in London, UK. Additionally, we aimed to explore HCPs views towards the diabetes education programmes available to patients with T2DM within these settings.

## METHODS
### Design
This qualitative cross-sectional study used semi-structured interviews with HCPs working in English primary care, secondary care and intermediate care services that served patients with T2DM from two inner city boroughs in North London.

This research took place in the context of a wider programme of work to develop, evaluate and implement a digital self-management programme for people with T2DM (HeLP-Diabetes). The research team conducted the interviews in this study as part of the HeLP-Diabetes implementation study (see refs [29 30] for more details).

### Setting
The setting was two densely populated urban boroughs in inner city London which were multi-ethnic and socially and economically diverse. The first borough has a population of 231 200 (based on 2017 estimates), with over a third born abroad and just under a half having a language other than English as their first language. The average household income (median modelled and 2012/3 figures) is £54 950 (England average £30 763). Just less than 5% of this population are unemployed and have no educational qualifications. One-third of children are reportedly living in poverty. For people aged 17+, 5.0% have diabetes. The second borough has a population of 242 500 people (based on 2017 estimates), just under half of whom were born abroad and have a language other than English as their first language. The average household income (median modelled and 2012/3 figures) is £67 990 (England average £30 763). Four per cent of the borough is unemployed and just under 2% of working age adults have no educational qualifications. A third of children are reportedly living in poverty. For people aged 17+, 3.9% have diabetes.[31]

At the time of the study there were four types of free education for people with T2DM provided in the boroughs (see table 1); Diabetes Education and Self-Management for Ongoing and Newly Diagnosed (DESMOND),[19] the Diabetes Self-Management Programme (previously referred to as Co-Creating Health),[32] the X-PERT Programme[33] and Healthy Living for People with T2DM (HeLP-Diabetes). HeLP-Diabetes is an online T2DM self-management programme which had been introduced to these boroughs by this research team as part of a wider

| Table 1 | Diabetes education available in the two boroughs | | | | |
|---------|---------|---------|---------|---------|---------|
| **Name** | **Delivery** | **Ethos** | **Duration** | **Target population** | **Access** |
| HeLP-Diabetes | Online | Online tool for adults with type 2 diabetes to learn knowledge and skills to manage their condition. The programme takes a holistic view of self-management and addresses a wide range of patient needs including medical management, emotional management and role management (such as adapting lifestyle or life roles). | Available 24/7 for as long as patient wants. | Type 2 diabetes | Referrals are made via health professional or self-referral. |
| DESMOND | Face-to-face group based | The programme teaches patients about diabetes and provides lifestyle advice so that they are better able to self-manage their condition. | One day. | Type 2 diabetes | Referrals are made via health professional. |
| Diabetes Self-Management Programme | Face-to-face group based | Aims to help participants strengthen their health-related behaviours. It does this by developing health literacy, building appreciation of peer support, developing collaborative decision-making skills and building knowledge of self-management techniques as well as participants' skills and confidence to use these techniques. | Runs over seven weekly sessions, lasting 3 hours per session. | Diabetes (types 1 and 2) | Referrals are made via health professional. |
| X-PERT Programme | Face-to-face group based | Aims to help patients cope with their health condition and improve their quality of life by learning new skills to manage their condition on a daily basis. | Six-week course. Each weekly session lasts two and a half hours. | Adults with one or more long-term health conditions (including diabetes). The course is also available for carers. | Self-referral or via health professional. |

DESMOND, Diabetes Education and Self-Management for Ongoing and Newly Diagnosed.

programme grant of research[34] and was much newer in the boroughs than the other forms of DSME.

In primary care settings, GPs have overall responsibility for the care and treatment of people with T2DM and provide referrals to specialist diabetes care. Diabetes specialist nurses are nurses with specialist knowledge of diabetes, providing support and advice and are often responsible for organising access to other specialists. They are usually based in secondary and tertiary care services. Practice nurses work in GP practices or diabetes clinics and assist with diabetes care. These three professional groups are the ones most likely to provide patients with referrals to DSME. Healthcare assistants (HCAs) work in GP practices and support patients with diabetes. In the two boroughs in this study, HCAs are trained in conducting T2DM self-management appointments with patients undertaking tasks such as assisting goal setting and action planning at diabetes reviews. Administrative staff and practice managers are mainly responsible for clerical tasks, however with the introduction of HeLP-Diabetes in the boroughs these staff were undertaking roles with patients such as registering patients to use the online education, assisting patients in accessing diabetes-related content and chasing up referrals to use the programme. Commissioning officers are responsible for decisions around commissioning diabetes services, including DSME and evaluating them. They also provide support to implement commissioned services and engage staff in delivering them.

## Sample
Sampling for HCPs interviews was purposive to capture the views from a range of HCPs working across the boroughs providing care to patients with T2DM. T2DM care was provided in primary care (37 GP practices in borough 1 and 32 GP practices in borough 2), community care (an intermediate diabetes service in each borough) and secondary care (three hospital trusts, two serving mainly patients from borough one and one serving patients mainly from borough 2). Twenty-six GPs, nurses, healthcare assistants, administrative staff, practice managers and commissioners were contacted via email and invited to take part in an interview throughout the duration of the study period (between July 2013 and August 2015).

## Data collection
Topic guides were developed with reference to previous research on self-management and DSME, with input from the wider project multidisciplinary steering group and

were informed by a theory of implementation (Normalisation Process Theory, NPT).[35 36]

NPT is widely used in process evaluations of innovations in healthcare organisation and delivery.[37 38] It focuses on the 'work' of implementation. This is represented by four constructs: Coherence: *what is the work that people do to understand and make sense of a practice*; Cognitive participation: *what is the work that people do to engage and support a new practice*; Collective action: *what is the work that people do to enact a new practice, and make it workable and integrate it in context*; and Reflexive monitoring: *what is the work that people do to reflect on and evaluate enacting a new practice in context*.

The topic guide was piloted with a member of the study team who was also a GP (this interview was excluded from the analysis). All interviews were semi-structured and conducted face-to-face by the same researcher (JR) who is an experienced female qualitative researcher who had worked in the boroughs implementing HeLP-Diabetes. All interview participants had been contacted before the day of interview to discuss the research and all participants provided informed consent. Most participants had met the researcher prior to the interviews in her role implementing HeLP-Diabetes and were aware of the research objectives of the wider programme grant.[34] All interviews were conducted in the HCPs' consultation rooms, or at the researcher's University (dependent on participant preference) and lasted between 30 min and an hour. Interviews were audio recorded and the researcher made field notes following each interview. Interviews continued until no new themes were apparent and thus representing data saturation (as described by Urquhart[39] and Given[40]).

## Data analysis

Data collection and analysis were conducted concurrently, with analysis starting as soon as interviews were transcribed. Corrected and anonymised transcripts were loaded into Nvivo 10 software[41] ready for coding. Although NPT had been used to inform the topic guide and ensure data on the relevant issues were collected, an inductive approach to analysis was employed to ensure the issues participants judged to be important were captured, as opposed to constraining their answers to the categories in NPT. This approach was also taken in analysis where initially an inductive approach to analysis was taken to capture responses, followed later by mapping the analysis onto the constructs of NPT (see figure 1 for the process of analysis). First, each transcript was read by JR and summaries of the main themes and impressions from each transcript were created to generate a feeling for each of the interviews and as a quick reference point for each interview. The themes that were identified in this initial analysis were discussed within the core team *(names removed for review purposes)*. To obtain other interpretations of the data from a range of perspectives, these themes were presented to the project's multidisciplinary steering group where the themes were discussed and refined. In addition to the steering group, a data clinic was held to

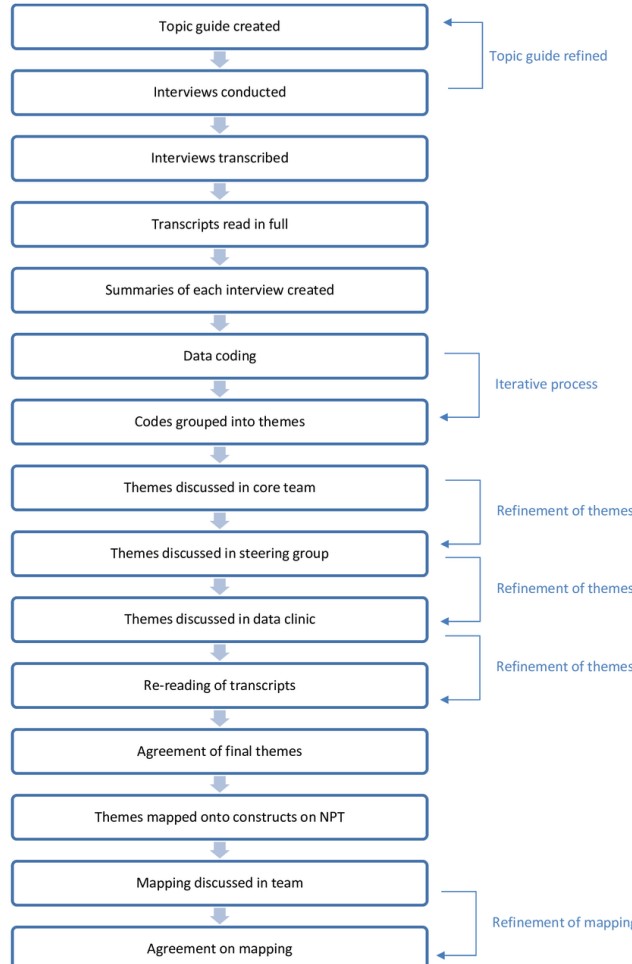

**Figure 1** Process of data analysis. NPT, Normalisation Process Theory.

explore the rigour and reliability of the themes from the initial analysis. Eight qualitative researchers from a range of disciplines (health services research, sociology, psychology and epidemiology) attended the data clinic and themes were discussed and refined and new themes were considered. Following the data clinic, all interview transcripts were re-read and recoded where appropriate; the themes were refined and additional ones created until a final set of themes emerged that were agreed on by all co-authors.

After agreement on themes had been reached, it was clear that the data resonated strongly with the constructs of NPT. In a second step to the analysis, themes were mapped onto constructs of NPT (Coherence, Cognitive Participation, Collective Action and Reflexive Monitoring) (see table 2). This required the researchers to re-read data within the themes and allocate the themes to appropriate constructs. This sometimes meant that the data coded under one theme was categorised into two or three different constructs. All themes could be applied to at least one construct. This approach has been used successfully in other research (eg,[38 42–45]) and provided

**Table 2** Mapping themes onto constructs of NPT

| Themes | Description of the theme | Construct of NPT |
|---|---|---|
| Perceptions of self-management | HCPs describe their views of the self-management approach to diabetes care being promoted within the service. | Coherence. |
| Barriers and facilitators to the self-management approach to diabetes | The difficulties to implementing a self-management approach with patients and the benefits of the approach. | Coherence, cognitive participation, collective action; reflexive monitoring. |
| HCPs and patient interactions | The way that HCPs and patient interactions are affected by self-management and DSME. | Collective action. |
| Perceptions of current DSME | HCPs views on group-based and online DSME. | Coherence. |
| HCPs role in promoting self-management and DSME | HCPs views about the extent and limitations of their roles in supporting patients to self-management and participate in DSME. | Coherence cognitive participation, Collective action. |
| Improving uptake of DSME | HCPs views on how patient participation in DSME could be increased. | Reflexive monitoring. |

DSME, diabetes self-management education; HCP, healthcare professional; NPT, Normalisation Process Theory.

confidence that the themes were data driven (although it is acknowledged that the use of the NPT to develop interview topic guides is likely to have affected the data collected), and meant that the robustness of NPT in explaining the data could be tested against the themes during this mapping process. The findings are presented within the NPT framework, together with illustrative quotations.

### Patient and public involvement

The wider programme grant, of which this study formed part of one work package, had significant patient and public involvement (PPI) input (both health professionals working in diabetes care and patients with T2DM) throughout (please see ref[46] for details). For this study specifically, two PPI representatives advised on the topic guide development and interpretation of findings. Both were invited to be part of developing the manuscript.

### RESULTS
### Characteristics of research participants

Twenty-two HCPs (of the 26 approached) took part in interviews, four HCPs did not respond to email requests to participate. The interview sample represented a diverse range of professional roles, experience in current role, ethnicities and experience with DSME (table 3). The sample worked within 11 different GP practices, both intermediate services, one hospital and a commissioning group.

### Coherence

It was clear that all participants were aware of the 'party line' on the importance of the self-management approach for patients with T2DM and for health services and that this policy view had become normalised within practice as the accepted approach for managing patients with T2DM. HCPs were knowledgeable about the intended benefits

and advantages of the self-management approach. The self-management approach was still viewed as fairly new, especially among HCPs who had been in their roles for

**Table 3** Participant characteristics

| | HCP (n=22) |
|---|---|
| Age | n (%) |
| 18–24 | 1 (4.5) |
| 25–34 | 3 (13.6) |
| 35–44 | 7 (31.8) |
| 45–54 | 6 (27.3) |
| 55–64 | 4 (18.2) |
| 65–74 | 1 (4.5) |
| Female, n (%) | 16 (72.7) |
| Role | n (%) |
| GP* | 4 (18.2) |
| Nurse† | 10 (45.5) |
| HCA | 3 (13.6) |
| Reception/admin | 3 (13.6) |
| Practice manager | 1 (4.5) |
| Commissioner | 1 (4.5) |
| Ethnicity | n (%) |
| White British | 16 (72.7) |
| South Asian | 3 (13.6) |
| Other | 3 (13.6) |
| Attended DSME taster session, n (%) | 15 (68.2) |
| Experience with online DSME, n (%) | 13 (59.1%) |

*Includes GP partners and salaried GPs.
†Includes practice nurses, diabetes specialist nurses and advanced nurse practitioners.
DSME, diabetes self-management education; HCA, Health care Assistant; HCP, healthcare professional.

many years, who viewed it as a new way of doing things. There was ambivalence in HCPs' descriptions about the value of this approach for certain patient groups. Some HCPs contested the value of self-management for patients such as those with mental health problems, learning difficulties, no or low education and/or health literacy, with HCPs suggesting that they should retain a duty of care for these patients.

> you still have to, if you like, retain responsibility as a professional a bit more for some of these hard to reach people... it's important not to throw the baby out with the bathwater, and say, oh, diabetes is all about the patients' responsibility (#11: Female GP partner. 17 years in role, experience of DSME taster session and online DSME)

In terms of DSME more specifically, although all HCPs were aware that there were DSME options to refer patients to, there was less coherence about what these were and what they entailed. Awareness was polarised between those who had attended or taught on a DSME course and those who had not. For those who had no personal experience with DSME many reported gaining their knowledge about what DSME consisted of and its value from patients who had attended.

> I haven't been to one, but from what patients have told me... I think they do...lifestyle changes (#4: Female practice nurse, 2 years in role, no experience of DSME taster session, experience of online DSME)

Because the online DSME was a new initiative being implemented within the service, awareness about it among HCPs was high. The value of group-based and online DSME were often presented in contrast to one another with the strength of one being the weakness of the other. For example, group-based education was seen as particularly beneficial in terms of the social and peer support that patients could gain from learning with other people with T2DM. Online DSME was presented as most beneficial for those people who would have difficulty in attending group-based DSME due to other commitments (work, caring responsibilities), those who disliked groups and, those who had different learning styles (ie, preferred to learn at their own pace and revisit information).

### Cognitive participation
Uncertainties over the legitimacy and value of self-management for some patients clearly impacted on HCPs' willingness to promote this approach to all. HCPs described impotence in trying to engage patients with self-management when faced with resistance from patients or when patients were not fulfilling the roles that self-management placed on them.

> if people don't want to make changes there is little that we can do. We can't force people (#16: Female diabetes specialist nurse, 1 year in role, experience of DSME taster session, no experience of online DSME)

Some HCPs expressed the view that patients should take more responsibility for their own care and that it was not a legitimate part of professionals' role to always be chasing patients to do things.

> I think they need to take more responsibility; quite often they don't turn up for their follow ups and, sort of, monitoring. (#4: Female practice nurse, 2 years in role, no experience of DSME taster session, experience of online DSME)

HCPs presented evidence of a tension when discussing benefits of self-management education. Often the benefit was framed in terms of being able to provide patients with education that they did not have the time to provide themselves due to a lack of resources and competing demands. However, there was the sense that, in an ideal world they would have preferred to keep the imparting of knowledge as part of their role and were frustrated that time pressures did not allow this. This gave rise to a sense of loss of control in terms of providing all the care needed for a patient. In some cases however (as described in cognitive participation), HCPs described holding onto this for certain patients for whom they did not feel self-management was appropriate.

> you don't have enough time, one-to-one, to do the information giving, which you do need to do and the self-management support. And so you can see that when people go on programmes, they come back so much better informed...So there's a bit of a frustration when you see people one-on-one. (#2 Female diabetes specialist nurse, 12 years in role, experience of DSME taster session, no experience of online DSME)

### Collective action
The role of many HCPs in promoting attendance was limited to providing a referral. There was little evidence that HCPs generally perceived their role to extend beyond this.

> We can only give them the form, I mean, there's no... I can't walk them up there. (#6 Male GP partner, 3 years in role, no experience of DSME taster session or online DSME)

However, there were exceptions to this, and some HCPs, particularly those who had direct experience with DSME, described it as a core part of their work to engage patients with the idea of self-management and to attend DSME, even if patients were reluctant.

> I think that you have to chip chip away, build a relationship, you know, and try to gradually keep them onboard (#11: Female GP partner. 17 years in role, experience of DSME taster session and online DSME)

There was also a sense that relationships between HCPs within practices were important for how far self-management and DSME could be implemented. One nurse described how it had not been easy to embed work to

support patient self-management within the practice. She had been trying to get her practice to agree to take on the online DSME to support patients with self-management but had experienced difficulty because the lead GP at the practice did not buy into the idea of it; she spoke about having to try and embed practices on her own, without support which had made it not possible to implement the DSME.

> I kind of have to hoe my own row. He's not obstructive, but he's got a very clear idea of what he thinks is important, and what isn't. It's not always easy. (#20: Female practice nurse, 17 years in role, experience of DSME taster session, no experience of online DSME)

Other HCPs reported that they perceived resistance around putting a self-management approach into action within their practice because of the increased workload that this approach was perceived to create. There was frustration expressed that the new emphasis on the self-management approach from the service was creating many additional tasks which they were being asked to absorb into an already overwhelming workload, without any tasks being removed. A paradox arose whereby the ideal of patient self-management could not be achieved by patients on their own and instead requires work on the part of the health professional, as described by one GP:

> Yes, they would love their patients to self-manage so why don't they just go off and do it, and the thought of starting digging actually creates much more work… That's the other thing of, well, self-management takes time and we've been asked to do lots and lots. So I'm being asked to do more…what are we going to stop doing? (#19: Male salaried GP, 1 year in role, experience of DSME taster session and online DSME)

For group-based DSME, HCPs spoke of adhering to guidelines and giving referrals to all newly diagnosed patients. This was despite nearly all HCPs describing patients they did not think would benefit from attending.

> Not everyone will be suitable. I think I've had a couple of patients where they would have liked to have taken a relative, and … one lady who had anxiety… she could only stay for the half-day. (#9 Female practice nurse, 4 years in role, experience of DSME taster session and online DSME)

However, in terms of online DSME, HCPs were more able to implement their own criteria for assessing who would be suitable to participate (presumably because this service was not incentivised at a practice level through the QOF, as group DSME was) which resulted in referrals being withheld from patients for whom it was not deemed appropriate.

> I normally say to them… do you feel comfortable…, using a computer?…If they say, no, I'm not interested, then I don't take it any further. (#8 Male receptionist/

administrator, 1 year in role, no experience of DSME taster session, experience of online DSME)

Although it was clear that self-management was the approach being promoted within general practice, this was not always translated into action on the part of the patients, who in many cases, in the opinion of HCPs, preferred care to remain in the hands of the health professional, thus creating a tension between the approach HCPs were expected to promote and the needs and preferences of patients.

> most of them they don't want to look at their blood results, they'd rather go through it with us and that's probably just because that's what they're used to. (#4: Female practice nurse, 2 years in role, no experience of DSME taster session, experience of online DSME)

The work of encouraging patients to manage their condition was sometimes described as frustrating or resulted in HCPs feeling that they were nagging patients. Others described how this work forced them into roles that they were uncomfortable with assuming, such as that of a detective.

> And so a lot of the time it's like being a detective … you know that it's [poor control]) about something that they're doing at home that they're not sharing with you (#2: Female diabetes specialist nurse, 12 years in role, experience of DSME taster session, no experience of online DSME)

Despite HCPs describing engaging in many different types of work to support patient self-management, there were only a few examples described of work to encourage patients to take up offers of participating in education more specifically. GPs, although the ones who were often responsible for providing referrals to DSME, were rarely the HCPs group to do the work of following up non-attenders. This often fell to healthcare assistants or reception staff who obtained information from the courses or online intervention on patients who did not participate; they then followed them up with additional offers via mail. Nurses also reported in engaging in some discussions with patients about attendance.

> The doctors obviously offer it to the patients (DSME) …if they've still not signed up when I send out the result letters, I just put a little reminder in that there is this website called Help-Diabetes, so I sort of try and get as many patients as we can. (#7: Female healthcare assistant, 9 years in role, no experience of DSME taster session, experience of online DSME)

### Reflexive monitoring

HCPs did not report in engaging regularly in any activities which would allow them to reflect on DSME. There were no formal systems in place for monitoring patient attendance. This appeared to be a haphazard process and varied by primary care practice. Some HCPs were aware

that monitoring did take place but were unaware of any data relating to the number of patients who participated in education. Despite this, HCPs did seem to be aware that the number of patients participating in education was low.

> I don't know what the uptake figures are like. I would imagine not very good. (#6: Male GP partner, 3 years in role, no experience of DSME taster session or online DSME)

They also reflected on this and suggested ways in which participation could be improved. The main way that HCPs thought they could engage more patients with DSME was by being able to offer different formats of DSME. It was widely acknowledged that no one approach would be suitable for all patients, and therefore it was a real strength for these boroughs having alternatives to group DSME available (such as the online DSME).

> I think if you're going to do self-management courses you have to have a menu of options, because there's nothing, you know, one size doesn't fit all so there's no way that everybody's going to want to do (group DSME). (#3: Male commissioner, experience of DSME taster session, no experience of online DSME)

HCPs discussed several strategies that could improve patient uptake of DSME. Several HCPs mentioned that patients should be able to attend group-based DSME directly after diagnosis, otherwise there is the risk, if patients have to join waiting lists, that they lose the impetus. Others recognised the role of HCPs in promoting available education and suggested that, to get patients more engaged, HCPs have to become more engaged.

> it's how the referrer is selling it, whether is he only selling it, or the GPs who are selling any form of structured education. If they're not explaining and all they've done is tick a box referral, then that's when the patients don't turn up. (#22: Female, diabetes specialist nurse, 10 years in role, experience of DSME taster session, no experience of online DSME)

Advertising DSME in the community was suggested as a way to raise patient awareness of education. Suggestions for locations for this included: pharmacies, older adult centres, supermarkets, television, national newspapers and libraries. For the online education specifically, HCPs thought that this could be advertised more widely using other online resources. There were also several suggestions that group-based and online education should advertise each other.

> if it's in the news or kind of just like an advert on TV or something it just kind of brings it to the attention of someone to think…it's kind of…got like a seal of approval…if you're told something just by one person you think well, no one else has told me (#18: Female receptionist/administrator, 7 years in role, no experience of DSME taster session or online DSME)

HCPs were concerned that neither group nor online DSME were available in languages other than English which excluded many patients who did not have English as a first language (although T2DM education DVDs were available in other languages). The timing and commitment required to attend the group-based courses were mentioned by many HCPs. It was suggested that running the courses at weekends or evenings could make it easier for more patients to attend. A few HCPs reported that patients might find it helpful to take a friend or relative to group education for support. Some HCPs suggested that more feedback from patients who had taken part in group or online DSME would help them to promote these services to other patients.

> it would be interesting to know… how they find it, and then I could say to other patients, well, actually patients have found this really helpful, you know. (#5: Female healthcare assistant, 10 years in role, no experience of DSME taster session, experience of online DSME)

## DISCUSSION

The study findings suggest that HCPs views towards self-management as a way of managing T2DM are ambiguous. Self-management had become the dominant approach for managing patients with T2DM in the study boroughs; this was described as arising out of necessity as opposed to investment by HCPs. Many HCPs describe this approach as valuable in principle, although many were concerned that for a proportion of patients, self-management is either clinically inappropriate or insufficient to support effective behaviour change (low coherence). There were also tensions about perceived responsibility for T2DM care. Several HCPs wanted to retain responsibility of care for certain patients for whom they did not feel self-management was appropriate. Other HCPs expressed frustration with patients and believed patients should take on more responsibility for looking after their T2DM. There was less evidence of HCPs believing that engaging patients with DSME more specifically was a legitimate part of their role (low cognitive participation). GPs viewed their role as limited to providing referrals to this education. As such there was little evidence of collective action around following up referrals or checking that patients had attended DSME. A 'care pathway' for patients to attend education was not evident, and there were no accounts of relationships between primary care and the providers of DSME which are likely to be necessary to increase patient participation. Lack of formal systems to monitor attendance also impacted on HCPs ability appraise DSME and likely impacted efforts to promote it to patients.

This study adds to the existing evidence on the perceptions of HCPs towards DSME and their role in patient engagement and attendance. Whereas previous studies have reported that HCPs are very knowledgeable about

DSME,[24] this study found varying degrees of knowledge. If HCPs are to take on responsibility for promoting DSME to patients, they must be aware of what it is and its potential benefits. The HCPs in this study had concerns over the self-management paradigm more broadly which is consistent with other findings.[2 27 47] This study has found that despite the current focus on patient self-management by policy-makers, HCPs believe there are certain patients for whom this approach to diabetes care is not deemed suitable.

Using NPT has provided a framework by which to explain the findings from this study and has helped to develop implications and recommendations. Because of an ambiguity towards the benefits of DSME by HCPs (low coherence), providing HCPs with opportunities to gain personal experience with DSME, for example, through a 'taster session' or more education for HCPs by providers, is likely to increase coherence and perceived value. There was little evidence of collective action between HCPs and providers of DSME in promoting patient attendance. Future research could focus on establishing a better pathway between those providing group DSME and the HCPs recommending it and incorporating DSME into routine practice by, for example, practice based events and training. This could increase opportunities for HCPs to offer additional sessions to patients who did not attend, or offer alternative formats of education (eg, online). In addition, providing HCPs with feedback from DSME graduates might provide more opportunities for HCPs to appraise DSME and be useful for HCP promotion to other patients, and for their own perceptions of the benefit.

There were several 'hard-to-reach' groups identified by HCPs as not suitable to attend DSME, including those with mental health problems, low literacy and from non-English speaking backgrounds. Designing different types of courses for different groups of people that HCPs can refer patients to may increase opportunities for participation as well as promote positive perceptions on the suitability of DSME in HCPs. In one of the study boroughs, since the completion of this study, a diabetes education course has been created in Bengali. However, given the multiple languages spoken in these boroughs, even the commissioning of DSME in several other languages still leaves many unable to participate. There are also DSME programmes for adults with intellectual and developmental disabilities being developed and evaluated,[48] but these were not available in the boroughs at the time of the study.

Given previous work on the importance of DSME being 'sold' to patients by HCPs, and the findings from this study that often the work around getting patients to attend DSME is limited to providing a referral, helping HCPs to market DSME more effectively might be an important way to increase participation. Further work under way by this research team is exploring how conversations about self-management and DSME are conducted in healthcare settings and how these can be improved upon.[49]

The strengths of this study include the wide range of HCPs interviewed including GPs, nurses, commissioners, healthcare assistants and administrative staff, with varying degrees of knowledge and experience of DSME. However, the sample was limited to HCPs within two London boroughs, and therefore their views and the range of services available to patients may not generalise more widely, especially as these boroughs are particularly diverse in terms of ethnicity, culture and socioeconomic status. It is important to note that these interviews were conducted in the context of a wider study that was implementing the online DSME in the two study boroughs. The researcher conducting the interviews had also been involved in implementation activities within these National Health Service services and was known to many of the participants. This has the potential to have elicited socially desirable responses to questions about DSME. In addition, three members of the steering group for this study worked within these boroughs at the time and this might have resulted in a social desirability bias in responses. However, before interviews, participants were informed that the findings would be used to develop and improve HeLP-Diabetes and the way that it was offered to patients; thus giving participants' permission to be critical or negative. Indeed, many participants' were very forthcoming about their non-engagement with HeLP-Diabetes, self-management and DSME more generally, suggesting that participants felt comfortable giving honest accounts.

**Acknowledgements** We thank the all the research participants and the wider study steering group.

**Contributors** JR, FS, CD, KP, CM, SM. LY and EM all contributed to the design and analysis of the study. JR undertook data collection. JR wrote the first draft of the paper; all authors commented on this draft and approved the final version.

**Funding** This report presents independent research funded by the NIHR under its Programme Grants for Applied Research Programme (Grant Reference Number RP-PG-0609-10135). JR is funded by the National Institute of Health Research School for Primary Care Research (NIHR SPCR) Capacity Building Award 7 programme [UCL Award 7,172727].

**Competing interests** EM is the managing director of HeLP Digital, a not-for-profit Community Interest Company that disseminates digital health interventions to the NHS. She has not, and will not, receive any remuneration for this work. KP has worked with HeLP Digital, a not-for-profit Community Interest Company that disseminates digital health interventions to the NHS. He has not, and will not, receive any remuneration for this work.

**Patient consent for publication** Not required.

**Ethics approval** Ethical approval was gained from NRES Committee East Midlands–Leicester.

**Provenance and peer review** Not commissioned; externally peer reviewed.

**Data sharing statement** No additional data are available.

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
