## [Reviewer comments · BMJ Open]

ARTICLE DETAILS

TITLE (PROVISIONAL)	Health care professionals' views towards self-management and self-management education for people with type 2 diabetes
AUTHORS	Ross, Jamie; Stevenson, Fiona; Dack, Charlotte; Pal, Kingshuk; May, Carl; Michie, Susan; Yardley, Lucy; Murray, Elizabeth

VERSION 1 – REVIEW

REVIEWER	Ping Yein Lee Universiti Putra Malaysia
REVIEW RETURNED	06-Mar-2019

GENERAL COMMENTS	Review: Health care professionals' views towards self-management and self-management education for people with type 2 diabetes - This manuscript was a pleasure to read. It was well written, clear and succinct. The title and abstract give an accurate summary of the aim and focus of the study and the major findings. The study design was appropriate to explore the views of HCPs towards self-management and self-management education programme. There are some suggestions for improvement as highlighted in my comments below. Introduction: - The introduction is very well written. It highlights the gaps of literatures to justify the study. - Some suggestion for improvement: For international readers who are not familiar with the UK health system, it would be good to provide some background information of the specific role of GPs, nurses, health care assistants, receptionists and commissioners of care in diabetes self-management of patients with diabetes. Methods: - The method section provided a good overview of the study setting and the available diabetes self-management education programmes. - This section also provided clear description of topic guide development, data collection process and data analysis following standard criteria of reporting of qualitative research. Additional information as suggested below may be useful: o Twenty two HCPs (of the 26 approached) took part in interviews .Reasons for not participating?o Was anyone else present during the interviews besides the participants and researcher? Results and Discussion:
--

	- For each quote, would be good to provide some information about the health professionals' years of experience, whether they had attended DSME taster session and experience with online DSME. - This quote "I kind of have to hoe my own row. He's not obstructive, but he's got a very clear idea of what he thinks is important, and what isn't. It's not always easy." does not reflect the description in the text that well, may be adding information about the context or some quote before this statement will be helpful. - In term of interpretation of the HCPs not so much involve in promoting DSME or follow-up, it may be apply to GPs only? This may be due to different specific role play by different category of HCPs (doctor play the role of referring? nurses conducting the DSME and the care assistant did the follow up and tracing the defaulter?) in the system of which this may have contributed to certain issue highlighted by certain category of HCP? Putting this more clearly in the results and discussion will give better understanding of the underlying problem in the system in the implementation of DSME.
--	--

REVIEWER	Sophie Harris King's college London
REVIEW RETURNED	27-Mar-2019

GENERAL COMMENTS	This is an interesting piece of work and very valuable. I have split into major and minor revisions: The title and research question suggests HCP opinion, however 5 of those interviewed are not HCPs. The data needs to be reanalysed using the HCP only transcripts, themes identified and saturation reassessed and addressed as appropriate. Minor: Given that one of the findings was a lack of understanding of DSME there needs to be a little more about this in the introduction to outline All references to DSME barriers relate to T2DM as does the discussion & intro. It would be worth making it clear from the outset that this is specifically looking at T2DM and all reference to 'diabetes' is to T2DM. The introduction is based predominantly on patient reasons for non-attendance. The data presented is about attendance also. This study is addressing the referrer barriers and therefore the introduction should be highlighting the gap between referral and attendance, and the need to investigate the inability to translate referral to attendance. It may also be of value to highlight the effect that incentives have had on the referral rates and any other drivers that have influenced referrals. The methods are too long - page 8 2nd para didn't add anything, 3rd para (describing types of DSME) might be better placed in intro. role management is not a term that would be widely understood in this context There is no data as to why 4 people didn't accept invitation to interview FEMALE researcher is irrelevant This study is carried out by the same team that have been implementing the digital DSME. This carries a large research bias.
---

	This is not discussed at all in the study. There is no recognition of this being a problem, or discussion as to how this might have been overcome/addressed. There is no evidence for how data saturation occurred. The themes overlap and there isn't sufficient evidence of how these have been coded separately - e.g. p15 line 23-30, p17 line 31-38, p20 line 36-39 - all about control of patient management decisions. Discussion needs expansion to make the most of this study and the utility of it for future implementation of DSME. There is very little suggesting future direction and how barriers can be overcome. There is little about limitations/bias/generalisability.
--	---

VERSION 1 – AUTHOR RESPONSE

Response to reviewers comments			
Manuscript ID bmjopen-2019-029961 entitled "Health care professionals' views towards self-management and self-management education for people with type 2 diabetes"			
Reviewer 1 Comments		Response	Location in text
1	The introduction is very well written. It highlights the gaps of literatures to justify the study.	Thank you	N/A
2	Some suggestion for improvement: For international readers who are not familiar with the UK health system, it would be good to provide some background information of the specific role of GPs, nurses, health care assistants, receptionists and commissioners of care in diabetes self-management of patients with diabetes.	Thank you for this suggestion, we have included a summary of each of the staff roles in relation to diabetes care in this setting.	Pages 10-11 Lines 224-243
3	The method section provided a good overview of the study setting and the available diabetes self-management education programmes.	Thank you	N/A
4	This section also provided clear description of topic guide development, data collection process and data analysis following standard criteria of reporting of qualitative research.	Thank you	N/A
5	Twenty two HCPs (of the 26 approached) took part in interviews .Reasons for not participating?	We have now included the following: "4 HCPs did not respond to email requests to participate" As they never responded, we have no data on the reasons for not participating.	Page 16 lines 342-343
6	Was anyone else present during the interviews besides the participants and researcher?	This has been clarified in the text. No, no one else was	N/A

		present	
7	For each quote, would be good to provide some information about the health professionals' years of experience, whether they had attended DSME taster session and experience with online DSME.	Thank you for this excellent suggestion; We have provided this information for each quote	Throughout results section
8	This quote "I kind of have to hoe my own row. He's not obstructive, but he's got a very clear idea of what he thinks is important, and what isn't. It's not always easy." does not reflect the description in the text that well, may be adding information about the context or some quote before this statement will be helpful.	Thank you, we have now amended the text that accompanies this quote to provide clarity.	Page 20-21 Lines 454-462
9	In term of interpretation of the HCPs not so much involve in promoting DSME or follow-up, it may be apply to GPs only? This may be due to different specific role play by different category of HCPs (doctor play the role of referring? nurses conducting the DSME and the care assistant did the follow up and tracing the defaulter?) in the system of which this may have contributed to certain issue highlighted by certain category of HCP? Putting this more clearly in the results and discussion will give better understanding of the underlying problem in the system in the implementation of DSME.	Thank you for this comment. We have tried to make this clearer in the results where we discuss this. And have clarified in the discussion.	Page 23 line 532 Page 27 Line 633
	Reviewer 2 Comments	Response	Location in text
10	The title and research question suggests HCP opinion, however 5 of those interviewed are not HCPs. The data needs to be reanalysed using the HCP only transcripts, themes identified and saturation reassessed and addressed as appropriate.	Thank you for this comment. Although not clinical roles, the other professional groups we have represented in this paper are involved in the referral, delivery, support and implementation of DSME within these services, and in our view, should be considered as healthcare professionals. We therefore respectfully disagree with the suggestion to remove their opinions from the manuscript. We have provided a description of how these different staff groups play a role in diabetes care in the introduction to help the readers understand this more clearly, and thank you for pointing this potential confusion out.	Pages 10-11 Lines 223-242
11	Given that one of the findings was a lack of	Thank you, we think part of this	Page 6

	understanding of DSME there needs to be a little more about this in the introduction to outline	comment might be missing, however, we have included this point in the introduction.	Lines 146-148
12	All references to DSME barriers relate to T2DM as does the discussion & intro. It would be worth making it clear from the outset that this is specifically looking at T2DM and all reference to 'diabetes' is to T2DM.	Thank you, we have made reference to type 2 in the abstract and key points and have changed 'diabetes' to 'T2DM' where appropriate throughout the manuscript.	Throughout
13	The introduction is based predominantly on patient reasons for non-attendance. The data presented is about attendance also. This study is addressing the referrer barriers and therefore the introduction should be highlighting the gap between referral and attendance, and the need to investigate the inability to translate referral to attendance. It may also be of value to highlight the effect that incentives have had on the referral rates and any other drivers that have influenced referrals.	Thank you for pointing this out, we have now included a paragraph on this in the introduction.	Pages 5-6 Lines 126-132
14	The methods are too long - page 8 2nd para didn't add anything, 3rd para (describing types of DSME) might be better placed in intro.	Thank you, we have removed the paragraph you refer to.	Page 8
15	role management is not a term that would be widely understood in this context	Thank you, we have added in further clarification on what is meant by role management.	Table 1
16	There is no data as to why 4 people didn't accept invitation to interview	We have now included the following: "4 HCPs did not respond to email requests to participate"	Page 16 lines 342-343
17	FEMALE researcher is irrelevant	We have included this as the journals preferred checklist for reporting qualitative studies asks for this information to be reported (COREQ checklist)	Page 12 Line 274
18	This study is carried out by the same team that have been implementing the digital DSME. This carries a large research bias. This is not discussed at all in the study. There is no recognition of this being a problem, or discussion as to how this might have been overcome/addressed.	We do refer to this is already in the paper- please see page 9 (lines 216-219) and again on page 12 (lines 273-279) in the methods. In order to address your comment further we have added further reference to this in the methods and discussion.	Page 8 Lines 181-184 Page 30 Lines 695-707
19	There is no evidence for how data saturation occurred.	We state on page 11 that "Interviews continued until no	Page 13 Lines 282-

		new themes were apparent” and have added to this “and thus representing data saturation (as described by (35, 36))”. And provide references to qualitative literature that use the point in coding/analysis when no new codes/themes emerge as evidence of data saturation.	284
20	The themes overlap and there isn't sufficient evidence of how these have been coded separately - e.g. p15 line 23-30, p17 line 31-38, p20 line 36-39 - all about control of patient management decisions.	We describe in table 2 how our themes have been mapped onto constructs of NPT and this table, along with the text on page 14 (lines 315-317), acknowledges that some of the themes span more than one NPT construct. In the example you point out we coded this type of data to the theme 'HCPs role in promoting self-management and DSME' and this does in deed appear throughout the process of implementing self-management and DSME. As HCPs had concerns over the suitability of the self-management approach for some (coherence), this led them to question whether it was the right approach for all (cognitive participation) and in turn this affected the work that they enacted around promoting a self-management approach in all cases (collective action). We've tried to make this narrative clearer and have made a change to table 2.	Table 2 Page 19 line 422
21	Discussion needs expansion to make the most of this study and the utility of it for future implementation of DSME. There is very little suggesting future direction and how barriers can be overcome. There is little about limitations/bias/generalisability.	We have added to the discussion to include future directions, addressing identified barriers and the limitations/bias.	Page 29 Lines 668-687

22	Please include Figure legend/caption at the end of your main manuscript.	Thank you this is now included	Last page
23	Please provide better qualities figures, ensuring the figures are not pixelated when zoomed in on. Figures can be supplied in TIFF, JPG or PDF format (figures in DOCUMENT, EXCEL or POWERPOINT format will not be accepted), we also request that they have a resolution of at least 300 dpi and 90mm x 90mm of width	The figure has now been supplied as a PDF	Attachment

VERSION 2 – REVIEW

REVIEWER	Ping Yein Lee Universiti Putra Malaysia Malaysia
REVIEW RETURNED	07-Jun-2019
GENERAL COMMENTS	The authors have address all my comments in the revised manuscript